# Program Synthesis and Semantic Parsing
# with Learned Code Idioms

**Richard Shin**[*]
UC Berkeley
ricshin@berkeley.edu

**Miltiadis Allamanis, Marc Brockschmidt & Oleksandr Polozov**
Microsoft Research
{miallama,mabrocks,polozov}@microsoft.com

## Abstract

Program synthesis of general-purpose source code from natural language specifications is challenging due to the need to reason about high-level patterns in the target program and low-level implementation details at the same time. In this work, we present PATOIS, a system that allows a neural program synthesizer to explicitly interleave high-level and low-level reasoning at every generation step. It accomplishes this by automatically mining common *code idioms* from a given corpus, incorporating them into the underlying language for neural synthesis, and training a tree-based neural synthesizer to use these idioms during code generation. We evaluate PATOIS on two complex semantic parsing datasets and show that using learned code idioms improves the synthesizer's accuracy.

## 1 Introduction

Program synthesis is a task of translating an incomplete specification (*e.g.* natural language, input-output examples, or a combination of the two) into the most likely program that satisfies this specification in a given language [15]. In the last decade, it has advanced dramatically thanks to the novel neural and neuro-symbolic techniques [5, 10, 19], first mass-market applications [28], and massive datasets [9, 39, 41]. Table 1 shows a few examples of typical tasks of program synthesis from natural language. Most of the successful applications apply program synthesis to manually crafted domain-specific languages (DSLs) such as FlashFill and Karel, or to subsets of general-purpose functional languages such as SQL and Lisp. However, scaling program synthesis to real-life programs in a general-purpose language with complex control flow remains an open challenge.

We conjecture that one of the main current challenges of synthesizing a program is insufficient separation between high-level and low-level reasoning. In a typical program generation process, be it a neural model or a symbolic search, the program is generated in terms of its *syntax tokens*, which represent low-level implementation details of the latent high-level *patterns* in the program. In contrast, humans switch between high-level reasoning (*"a binary search over an array"*) and low-level implementation ("`while l < r: m = (l+r)/2 ...`") repeatedly when writing a single function. Reasoning over multiple abstraction levels at once complicates the generation task for a model.

This conjecture is supported by two key observations. First, recent work [12, 25] has achieved great results by splitting the synthesis process into *sketch generation* and *sketch completion*. The first stage generates a high-level sketch of the target program, and the second stage fills in missing details in the sketch. Such separation improves the accuracy of synthesis as compared to an equivalent end-to-end generation. However, it allows only one stage of high-level reasoning at the root level of the program, whereas **(a)** real-life programs involve common patterns at all syntactic levels, and **(b)** programmers often interleave high-level and low-level reasoning during implementation.

---

[*]Work done partly during an internship at Microsoft Research.

Table 1: Representative program synthesis tasks from real-world semantic parsing datasets.

| Dataset | Natural Language Specification | Program |
|---------|-------------------------------|---------|
| Hearthstone [24] | *Mana Wyrn (1, 3, 1, Minion, Mage, Common)* <br> *Whenever you cast a spell, gain +1 Attack.* | ```#... def create_minion(self, player): return Minion(1, 3, effects=[Effect( SpellCast(), ActionTag(Give( ChangeAttack(1), SelfSelector()))])``` |
| Spider [41] | *For each stadium, how many concerts are there?* <br><br> *Schema:* <br> *stadium = {stadium_id, name, ...}, ...* | ```SELECT T2.name, COUNT(*) FROM concert AS T1 JOIN stadium AS T2 ON T1.stadium_id = T2.stadium_id GROUP BY T1.stadium_id``` |

Second, many successful applications of inductive program synthesis such as FlashFill [14] rely on a manually designed DSL to make the underlying search process scalable. Such DSLs include high-level operators that implement common subroutines in a given domain. Thus, they **(i)** compress the search space, ensuring that every syntactically valid DSL program expresses some useful task, and **(ii)** enable logical reasoning over the domain-specific operator semantics, making the search efficient. However, DSL design is laborious and requires domain expertise. Recently, Ellis et al. [13] showed that such DSLs are learnable in the classic domains of inductive program synthesis; in this work, we target general-purpose code generation, where DSL design is difficult even for experts.

In this work, we present a system, called PATOIS, that equips a program synthesizer with automatically learned high-level *code idioms* (*i.e.* common program fragments) and trains it to use these idioms in program generation. While syntactic by definition, code idioms often represent useful semantic concepts. Moreover, they *compress* and *abstract* the programs by explicitly representing common patterns with unique tokens, thus simplifying generative process for the synthesis model.

PATOIS has three main components, illustrated in Figure 1. First, it employs nonparameteric Bayesian inference to mine the code idioms that frequently occur in a given corpus. Second, it marks the occurrences of these idioms in the training dataset as new named operators in an extended grammar. Finally, it trains a neural generative model to optionally emit these named idioms instead of the original code fragments, which allows it to learn idiom usage conditioned on a task specification. During generation, the model has the ability to emit entire idioms in a single step instead of multiple steps of program tree nodes comprising the idioms' definitions. As a result, PATOIS interleaves high-level idioms with low-level tokens at all levels of program synthesis, generalizing beyond fixed top-level sketch generation.

We evaluate PATOIS on two challenging semantic parsing datasets: Hearthstone [24], a dataset of small domain-specific Python programs, and Spider [41], a large dataset of SQL queries over various databases. We find that equipping the synthesizer with learned idioms improves its accuracy in generating programs that satisfy the task description.

## 2 Background

**Program Synthesis**   We consider the following formulation of the *program synthesis* problem. Assume an underlying programming language $\mathcal{L}$ of programs. Each program $P \in \mathcal{L}$ can be represented either as a sequence $y_1 \cdots y_{|P|}$ of its *tokens*, or, equivalently, as an *abstract syntax tree (AST)* $T$ parsed according to the context-free grammar (CFG) $\mathcal{G}$ of the language $\mathcal{L}$. The goal of a program synthesis model $f \colon \varphi \mapsto P$ is to generate a program $P$ that maximizes the conditional probability $\Pr(P \mid \varphi)$ *i.e.* the most likely program given the specification. We also assume a training set $\mathcal{D} = \{\langle \varphi_j, P_j \rangle\}_{j=1}^{|\mathcal{D}|}$, sampled from an unknown true distribution $\mathfrak{D}$, from which we wish to estimate the conditional probability $\Pr(P \mid \varphi)$.

In this work, we consider general-purpose programming languages $\mathcal{L}$ with a known context-free grammar $\mathcal{G}$ such as Python and SQL. Each *specification* $\varphi$ is represented as a *natural language task description*, *i.e.* a sequence of words $X = x_1 \cdots x_{|X|}$ (although the PATOIS synthesizer can be conditioned on any other type of incomplete spec). In principle, we do not impose any restrictions on the generative model $f$ apart from it being able to emit syntactically valid programs. However, as we detail in Section 4, the PATOIS framework is most easily implemented on top of *structural generative models* such as sequence-to-tree models [38] and graph neural networks [7, 21].

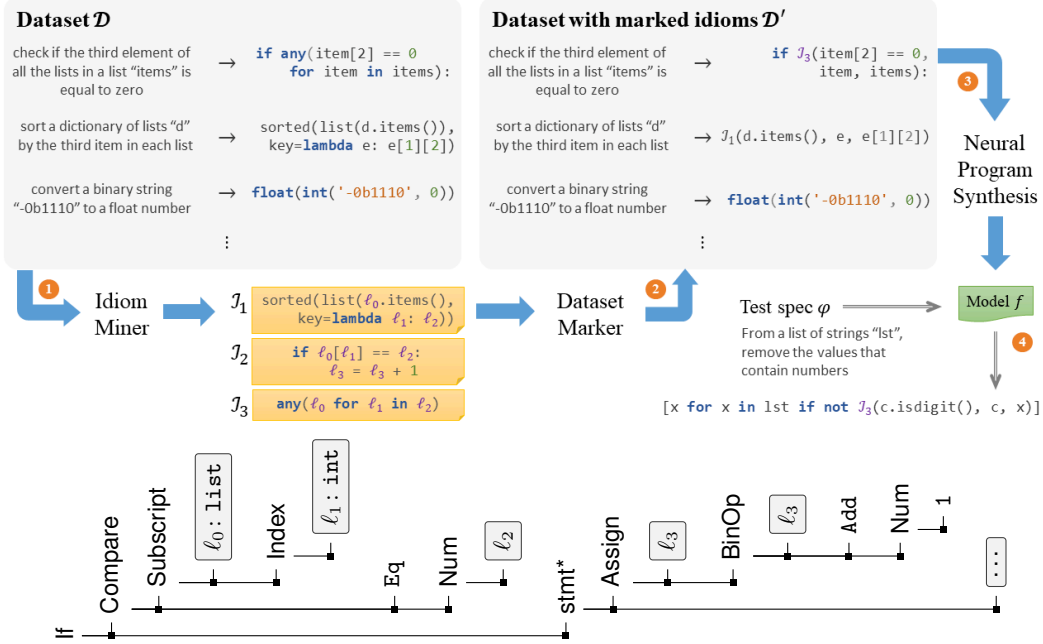

Figure 1: *Top:* An overview of PATOIS. A miner ① extracts common idioms from the programs in a given dataset. All the idiom occurrences in the dataset programs are ② marked as optional alternative grammar operators. The dataset with marked occurrences is used to ③ train a neural generative model. At inference time, the model ④ generates programs with named idioms, which are inlined before program execution. Note that idioms may have named subexpressions, may repeat, and may occur at any program level. For clarity, we typeset idioms using function-like syntax $\mathcal{I}_j(\ell_1, \ldots, \ell_k)$ in this paper, although they are actually represented as AST fragments with no syntax. *Bottom:* AST fragment representation of the idiom $\mathcal{I}_2$ in Python. Here sans-serif nodes are fixed non-terminals, monospaced nodes are fixed terminals, and boxed nodes are named arguments.

**Code Idioms**    Following Allamanis and Sutton [2], we define code idioms as *fragments* $\mathcal{I}$ of valid ASTs $T$ in the CFG $\mathcal{G}$, *i.e.* trees of nonterminals and terminals from $\mathcal{G}$ that may occur as subtrees of valid parse trees from $\mathcal{G}$. The grammar $\mathcal{G}$ extended with a set of idiom fragments forms a *tree substitution grammar* (TSG). We also associate a non-unique *label* $\ell$ with each nonterminal leaf in every idiom, and require that every instantiation of an idiom $\mathcal{I}$ must have its identically-labeled nonterminals instantiated to identical subtrees. This enables the role of idioms as *subroutines*, where labels act as "named arguments" in the "body" of an idiom. See Figure 1 for an example.

## 3  Mining Code Idioms

The first step of PATOIS is obtaining a set of frequent and useful AST fragments as code idioms. The trade-off between frequency and usefulness is crucial: it is trivial to mine *commonly occurring* short patterns, but they are often meaningless [1]. Instead, we employ and extend the methodology of Allamanis et al. [3] and frame idiom mining as a nonparameteric Bayesian problem.

We represent idiom mining as inference over *probabilistic tree substitution grammars* (pTSG). A pTSG is a probabilistic context-free grammar extended with production rules that expand to a whole AST fragment instead of a single level of symbols [8, 29]. The grammar $\mathcal{G}$ of our original language $\mathcal{L}$ induces a pTSG $\mathcal{G}_0$ with no fragment rules and with choice probabilities estimated from the corpus $\mathcal{D}$. To construct a pTSG corresponding to the extension of $\mathcal{L}$ with common tree fragments representing idioms, we define a distribution $\mathfrak{G}$ over pTSGs as follows.

We first choose a Pitman-Yor process [36] as a prior distribution $\mathfrak{G}_0$ over pTSGs. It is a nonparameteric process that has proven to be effective for mining code idioms in prior work thanks to its modeling of production choices as a Zipfian distribution (in other words, it implements the desired "rich get richer" effect, which encourages a smaller number of larger *and* more common idioms).

Formally, it is a "stick-breaking" process [31] that defines $\mathfrak{G}_0$ as a distribution for each *set of idioms* $\widetilde{\mathcal{I}}_N$ rooted at a nonterminal symbol $N$ as

$$\Pr(\mathcal{I} \in \widetilde{\mathcal{I}}_N) \stackrel{\text{def}}{=} \sum_{k=0}^{\infty} \pi_k \, \delta\left(\mathcal{I} = \mathcal{I}_k\right), \quad \mathcal{I}_k \sim \mathcal{G}_0$$

$$\pi_k \stackrel{\text{def}}{=} u_k \prod_{j=1}^{k-1} (1 - u_j), \ u_k \sim \text{Beta}\left(1 - d, \ \alpha + kd\right)$$

where $\delta(\cdot)$ is the delta function, and $\alpha$, $d$ are hyperparameters. See Allamanis et al. [3] for details.

PATOIS uses $\mathfrak{G}_0$ to compute a posterior distribution $\mathfrak{G}_1 = \Pr\left(\mathcal{G}_1 \mid T_1, \ldots, T_N\right)$ using Bayes' rule,

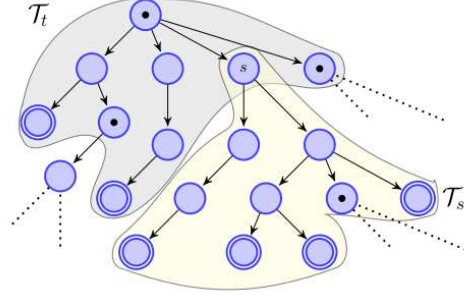

Figure 2: MCMC sampling for an AST (figure from [2]). Dots show the inferred nodes where the AST is split into fragments.

where $T_1, \ldots, T_N$ are concrete AST fragments in the training set $\mathcal{D}$. As this calculation is computationally intractable, we approximate it using type-based MCMC [23]. At each iteration $t$ of the MCMC process, PATOIS generates a pTSG $\mathcal{G}_t$ whose distribution approaches $\mathfrak{G}_1$ as $t \to \infty$. It works by sampling *splitting points* for each AST $T$ in the corpus $\mathcal{D}$, which by construction define a set of fragments constituting $\mathcal{G}_t$ (see Figure 2). The split probabilities of this Gibbs sampling are set in a way that incentivizes merging adjacent tree fragments that often cooccur in $\mathcal{D}$. The final idioms are then extracted from the pTSG obtained at the last MCMC iteration.

While the Pitman-Yor process helps avoid overfitting the idioms to $\mathcal{D}$, not all sampled idioms are useful for synthesis. Thus we *rank* and *filter* the idioms before using them in the training. In this work, we reuse two ranking functions defined by Allamanis et al. [3]:

$$\mathsf{Score}_{\mathsf{Cov}}\left(\mathcal{I}\right) \stackrel{\text{def}}{=} \text{coverage} = \mathsf{count}(T \in \mathcal{D} \mid \mathcal{I} \in T)$$

$$\mathsf{Score}_{\mathsf{CXE}}\left(\mathcal{I}\right) \stackrel{\text{def}}{=} \text{coverage} \cdot \text{cross-entropy gain} = \frac{\mathsf{count}(T \in \mathcal{D} \mid \mathcal{I} \in T)}{|\mathcal{D}|} \cdot \frac{1}{|\mathcal{I}|} \log \frac{\Pr_{\mathfrak{G}_1}(\mathcal{I})}{\Pr_{\mathfrak{G}_0}(\mathcal{I})}$$

and also filter out any *terminal* idioms (*i.e.* those that do not contain any named arguments $\ell$).

We conclude with a brief analysis of computational complexity of idiom mining. Every iteration of the MCMC sampling traverses the entire dataset $\mathcal{D}$ once to sample the random variables that define the splitting points in each AST. When run for $M$ iterations, the complexity of idiom mining is $\mathcal{O}(M \cdot \sum_{T \in \mathcal{D}} |T|)$. Idiom ranking adds an additional step with complexity $\mathcal{O}(|\widetilde{\mathcal{I}}| \log |\widetilde{\mathcal{I}}|)$ where $\widetilde{\mathcal{I}}$ is the set of idioms obtained at the last iteration. In our experiments (detailed in Section 5) we set $M = 10$, and the entire idiom mining takes less than 10 minutes on a dataset of $|\mathcal{D}| \approx 10{,}000$ ASTs.

## 4 Using Idioms in Program Synthesis

Given a set of common idioms $\widetilde{\mathcal{I}} = \{\mathcal{I}_1, \ldots, \mathcal{I}_N\}$ mined by PATOIS, we now aim to learn a synthesis model $f$ that emits whole idioms $\mathcal{I}_j$ as atomic actions instead of individual AST nodes that comprise $\mathcal{I}_j$. Achieving this involves two key challenges.

First, since idioms are represented as AST fragments without concrete syntax, PATOIS works best when the synthesis model $f$ is *structural*, *i.e.* it generates the program AST instead of its syntax. Prior work [7, 38, 40] also showed that tree- and graph-based code generation models outperform sequence-to-sequence models, and thus we adopt a similar architecture in this work.

Second, exposing the model $f$ to idiom usage patterns is not obvious. One approach could be to extend the grammar with new named operators $\mathsf{op}_{\mathcal{I}}(\ell_1, \ldots, \ell_k)$ for each idiom $\mathcal{I}$, replace every occurrence of $\mathcal{I}$ with $\mathsf{op}_{\mathcal{I}}$ in the data, and train the synthesizer on the rewritten dataset. However, this would not allow $f$ to learn from the idiom definitions (bodies). In addition, idiom occurrences often overlap, and any deterministic rewriting strategy would arbitrarily discard some occurrences from the corpus, thus limiting the model's exposure to idiom usage. In our experiments, we found that greedy rewriting discarded as many as $75\%$ potential idiom occurrences from the dataset. Therefore, a successful training strategy must preserve all occurrences and instead let the model *learn* a rewriting strategy that optimizes end-to-end synthesis accuracy.

To this end, we present a novel training setup for code generation that encourages the model to choose the most useful subset of idioms and the best representation of each program in terms of the idioms. It works by **(a)** marking occurrences of the idioms $\widetilde{\mathcal{I}}$ in the training set $\mathcal{D}$, **(b)** at training time, encouraging the model to emit *either* the whole idiom *or* its body for every potential idiom occurrence in the AST, and **(c)** at inference time, replacing the model's state after emitting an idiom $\mathcal{I}$ with the state the model would have if it had emitted $\mathcal{I}$'s body step by step.

## 4.1  Model Architecture

The synthesis model $f$ of PATOIS combines a *spec encoder* $f_{\text{enc}}$ and an *AST decoder* $f_{\text{dec}}$, following the formulation of Yin and Neubig [38]. The encoder $f_{\text{enc}}$ embeds the NL specification $X = x_1 \cdots x_n$ into word representations $\hat{X} = \hat{\boldsymbol{x}}_1 \cdots \hat{\boldsymbol{x}}_n$. The decoder $f_{\text{dec}}$ uses an LSTM to model the sequential generation of the AST in the depth-first order, wherein each timestep $t$ corresponds to *an action* $a_t$ — either (a) expanding a production from the grammar, (b) expanding an idiom, or (c) generating a terminal token. Thus, the probability of generating an AST $T$ given $\hat{X}$ is

$$\Pr(T \mid \hat{X}) = \prod_t \Pr(a_t \mid T_t, \hat{X}) \tag{1}$$

where $a_t$ is the action taken at timestep $t$, and $T_t$ is the partial AST generated before $t$. The probability $\Pr(a_t \mid T_t, \hat{X})$ is computed from the decoder's hidden state $\boldsymbol{h}_{t-1}$ depending on $a_t$.

**Production Actions**  For actions $a_t = \text{APPLYRULE}[R]$ corresponding to expanding production rules $R \in \mathcal{G}$ from the original CFG $\mathcal{G}$, we compute the probability $\Pr(a_t \mid T_t, \hat{X})$ by encoding the current partial AST structure similarly to Yin and Neubig [38]. Specifically, we compute the new hidden state as $\boldsymbol{h}_t = f_{\text{LSTM}} ([\boldsymbol{a}_{t-1} \parallel \boldsymbol{c}_t \parallel \boldsymbol{h}_{p_t} \parallel \boldsymbol{a}_{p_t} \parallel \boldsymbol{n}_{f_t}], \boldsymbol{h}_{t-1})$ where $\boldsymbol{a}_{t-1}$ is the embedding of the previous action, $\boldsymbol{c}_t$ is the result of soft attention applied to the spec embeddings $\hat{X}$ as per Bahdanau et al. [4], $p_t$ is the timestep corresponding to expanding the parent AST node of the current node, and $\boldsymbol{n}_{f_t}$ is the embedding of the current node type. The hidden state $\boldsymbol{h}_t$ is then used to compute probabilities of the syntactically appropriate production rules $R \in \mathcal{G}$:

$$\Pr(a_t = \text{APPLYRULE}[R] \mid T_t, \hat{X}) = \text{softmax}_R \left( g(\boldsymbol{h}_t) \right) \tag{2}$$

where $g(\cdot)$ is a 2-layer MLP with a $\tanh$ non-linearity.

**Terminal Actions**  For actions $a_t = \text{GETTOKEN}[y]$, we compute the probability $\Pr(a_t \mid T_t, \hat{X})$ by combining a small vocabulary $\mathcal{V}$ of tokens commonly observed in the training data with a *copying mechanism* [24, 30] over the input $X$ to handle UNK tokens. Specifically, we learn two functions $p_{\text{gen}}(\boldsymbol{h}_t)$ and $p_{\text{copy}}(\boldsymbol{h}_t, X)$ such that $p_{\text{gen}}$ produces a score for each vocabulary token $y \in \mathcal{V}$ and $p_{\text{copy}}$ computes a score for copying the token $y$ from the input. The scores are then normalized across the entries corresponding to the same constant, as in [7, 38].

## 4.2  Training to Emit Idioms

As discussed earlier, training the model to emit idioms presents computational and learning challenges. Ideally, we would like to extend Eq. (1) to maximize

$$\mathcal{J} = \sum_{\tau \in \mathcal{T}} \prod_{i=1}^{|\tau|} \Pr(a_{\tau_i} \mid T_{\tau_i}, \hat{X}) \tag{3}$$

where $\mathcal{T}$ is a set of different *action traces* that may produce the output AST $T$. The traces $\tau \in \mathcal{T}$ differ only in their possible choices of idiom actions $\text{APPLYRULE}[\text{op}_{\mathcal{I}}]$ that emit some tree fragments of $T$ in a single step. However, computing Eq. (3) is intractable because idiom occurrences overlap and cause combinatorial explosion in the number of traces $\mathcal{T}$. Instead, we apply Jensen's inequality and maximize a lower bound:

$$\log \mathcal{J} = \log \sum_{\tau \in \mathcal{T}} \prod_{i=1}^{|\tau|} \Pr(a_{\tau_i} \mid T_{\tau_i}, \hat{X}) \geq \log(|\mathcal{T}|) + \frac{1}{|\mathcal{T}|} \sum_{\tau \in \mathcal{T}} \sum_{i=1}^{|\tau|} \log \Pr(a_{\tau_i} \mid T_{\tau_i}, \hat{X}) \tag{4}$$

Let $A(T_t) = \{a_t^*\} \cup I(T_t)$ be the set of all valid actions to expand the AST $T_t$ at timestep $t$. Here $a_t^*$ is the action from the *original action trace* that generates $T$ using the original CFG and $I(T_t)$ is the set of idiom actions APPLYRULE[$\mathsf{op}_\mathcal{I}$] also applicable at the node to be expanded in $T_t$. Let $c(\mathcal{T}, t)$ also denote the number of traces $\tau \in \mathcal{T}$ that admit an action choice for the AST $T_t$ from the original action trace. Since each action $a \in A(T_t)$ occurs in the sum in Eq. (4) with probability $c(\mathcal{T}, t) \,/\, |A(T_t)|$, we can rearrange this sum over traces as a sum over timesteps of the original trace:

$$\frac{1}{|\mathcal{T}|} \sum_{\tau \in \mathcal{T}} \sum_{i=1}^{|\tau|} \log \Pr(a_{\tau_i} \mid T_{\tau_i}, \hat{X}) = \frac{1}{|\mathcal{T}|} \sum_t \sum_{a \in A(T_t)} \frac{c(\mathcal{T}, t)}{|A(T_t)|} \log \Pr(a \mid T_{\tau_i}, \hat{X})$$

$$= \sum_t \frac{1}{|A(T_t)|} \sum_{a \in A(T_t)} \frac{c(\mathcal{T}, t)}{|\mathcal{T}|} \log \Pr(a \mid T_{\tau_i}, \hat{X}) = \mathop{\mathbb{E}}_{T_t \sim \mathcal{T}} \frac{1}{|A(T_t)|} \sum_{a \in A(T_t)} \log \Pr(a \mid T_{\tau_i}, \hat{X})$$

$$\approx \sum_t \frac{1}{|A(T_t)|} \Big[ \log \Pr(a_t^* \mid T_t, \hat{X}) + \sum_{\mathcal{I} \in M(T_t)} \log \Pr(a_t = \text{APPLYRULE}[\mathsf{op}_\mathcal{I}] \mid T_t, \hat{X}) \Big] \qquad (5)$$

In the last step of Equation (5), we approximate the expectation over ASTs randomly drawn from all traces $\mathcal{T}$ using only the original trace (containing all possible $T_t$) as a Monte Carlo estimate.

Intuitively, at each timestep during training we encourage the model to emit *either* the original AST action for this timestep *or* any applicable idiom that matches the AST at this step, with no penalty to either choice. However, to avoid the combinatorial explosion, we only teacher-force the original generation trace (*not* the idiom bodies), thus optimizing the bound in Eq. (5). Figure 3 illustrates this optimization process on an example.

At inference time, whenever the model emits an APPLYRULE[$\mathsf{op}_\mathcal{I}$] action, we teacher-force the body of $\mathcal{I}$ by substituting the embedding of the previous action $\boldsymbol{a}_{t-1}$ with embedding of the *previous action in the idiom definition*, thus emulating the tree fragment expansion. Outside the bounds of $\mathcal{I}$ (*i.e.* within the hole subtrees of $\mathcal{I}$) we use the actual $\boldsymbol{a}_{t-1}$ as usual.

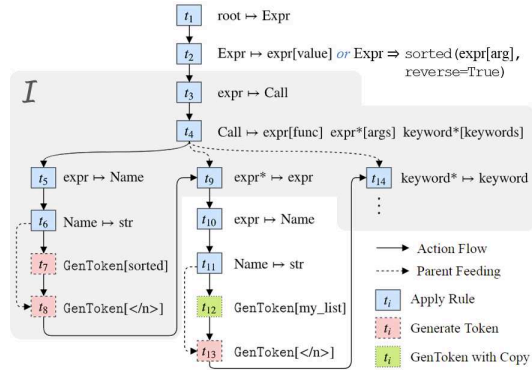

Figure 3: Decoding the AST `sorted(my_list, reverse=True)`, figure adapted from [38]. Suppose an idiom $\mathcal{I} = \texttt{sorted}(\boxed{\ell}, \texttt{reverse=True})$ is mined and added as an operator $\mathsf{op}_\mathcal{I}(\ell)$ to the grammar. At training time, PATOIS adjusts the cross-entropy objective at timestep $t_2$ to additionally allow $\mathsf{op}_\mathcal{I}$ as a valid production, with no change to further decoding. At inference time, if decoder emits an action $a_{t_2} = \text{APPLYRULE}[\mathsf{op}_\mathcal{I}]$, PATOIS unrolls $\mathcal{I}$ on the fly by teacher-forcing the shaded portion of the AST generation.

## 5 Evaluation

**Datasets**  We evaluate PATOIS on two semantic parsing datasets: Hearthstone [24] and Spider [41].

Hearthstone is a dataset of 665 card descriptions from the trading card game of the same name, along with the implementations of their effects in Python using the game APIs. The descriptions act as NL specs $X$, and are on average 39.1 words long.

Spider is a dataset of 10,181 questions describing 5,693 unique SQL queries over 200 databases with multiple tables each. Each question pertains to a particular database, whose schema is given to the synthesizer. Database schemas do not overlap between the train and test splits, thus challenging the model to generalize across different domains. The questions are on average 13 words long and databases have on average 27.6 columns and 8.8 foreign keys.

**Implementation**  We mine the idioms using the training split of each dataset. Thus PATOIS cannot indirectly overfit to the test set by learning its idioms, but it also cannot generalize beyond the idioms that occur in the training set. We run type-based MCMC (Section 3) for 10 iterations with $\alpha = 5$

Table 2: Ablation tests on the Hearthstone dev set.

| Model | K | Exact match | Sentence BLEU | Corpus BLEU |
|---|---|---|---|---|
| Baseline decoder | — | **0.197** | 0.767 | 0.763 |
| PATOIS, Score$_{Cov}$ | 10 | 0.151 | 0.781 | **0.785** |
| | 20 | 0.091 | 0.745 | 0.745 |
| | 40 | 0.167 | 0.765 | 0.764 |
| | 80 | **0.197** | 0.780 | 0.774 |
| PATOIS, Score$_{CXE}$ | 10 | 0.151 | 0.780 | 0.783 |
| | 20 | 0.167 | **0.787** | 0.782 |
| | 40 | 0.182 | 0.773 | 0.770 |
| | 80 | 0.151 | 0.771 | 0.768 |

Table 3: Ablation tests on the Spider dev set.

| Model | K | Exact match |
|---|---|---|
| Baseline decoder | — | 0.395 |
| PATOIS, Score$_{Cov}$ | 10 | 0.394 |
| | 20 | 0.379 |
| | 40 | 0.395 |
| | 80 | 0.407 |
| PATOIS, Score$_{CXE}$ | 10 | 0.368 |
| | 20 | 0.382 |
| | 40 | 0.387 |
| | 80 | **0.416** |

and $d = 0.5$. After ranking (with either Score$_{COV}$ or Score$_{CXE}$) and filtering, we use $K$ top-ranked idioms to train the generative model. We ran ablation experiments with $K \in \{10, 20, 40, 80\}$.

As described in Section 4, for all our experiments we used a tree-based decoder with a pointer mechanism as the synthesizer $f$, which we implemented in PyTorch [27]. For the Hearthstone dataset, we use a bidirectional LSTM [16] to implement the description encoder $\hat{X} = f_{\text{enc}}(X)$, similarly to Yin and Neubig [38]. The word embeddings $\hat{x}$ and hidden LSTM states $h$ have dimension 256. The models are trained using the Adadelta optimizer [42] with learning rate 1.0, $\rho = 0.95$, $\varepsilon = 10^{-6}$ for up to 2,600 steps with a batch size of 10.

For the Spider dataset, word embeddings $\hat{x}$ have dimension 300, and hidden LSTM states $h$ have dimension 256. The models are trained using the Adam optimizer [20] with $\beta_1 = 0.9$, $\beta_2 = 0.999$, $\varepsilon = 10^{-9}$ for up to 40,000 steps with a batch size of 10. The learning rate warms up linearly up to $2.5 \times 10^{-4}$ during the first 2,000 steps, and then decays polynomially by $(1 - t/T)^{-0.5}$ where $T$ is the total number of steps. Each model configuration is trained on one NVIDIA GTX 1080 Ti GPU.

The Spider tasks additionally include the *database schema* as an input in the description. We follow a recent approach of embedding the schema using relation-aware self-attention within the encoder [34]. Specifically, we initialize a representation for each column, table, and word in the question, and then update these representations using 4 layers of relation-aware self-attention [32] using a graph that describes the relations between columns and tables in the schema. See Section A in the appendix for more details about the Spider schema encoder.

## 5.1 Experimental Results

In each configuration, we compare the performance of equivalent trained models on the same dataset with and without idiom-based training of PATOIS. For fairness, we show the performance of the same decoder implementation described in Section 4.1 as a baseline rather than the state-of-the-art results achieved by different approaches from the literature. Thus, our baseline is the decoder described in Section 4.1 trained with a regular cross-entropy objective rather than the PATOIS objective in Equation (5). Following prior work, we evaluate program generation as a semantic parsing task, and measure **(i)** exact match accuracy and BLEU scores for Hearthstone and **(ii)** exact match accuracy of program sketches for Spider.

Tables 2 and 3 show our ablation analysis of different configurations of PATOIS on the Hearthstone and Spider dev sets, respectively. Table 4 shows the test set results of the best model configuration for Hearthstone (the test instances for the Spider dataset are unreleased). As the results show, small numbers of idioms do not significantly change the exact match accuracy but improve BLEU score, and $K = 80$ gives a significant improvement in both the exact match accuracy and BLEU scores. The improvement is even more pronounced on the test set with 4.5% improvement in exact match accuracy and more than 4 BLEU points, which shows that mined training set idioms

Table 4: Test set results on Hearthstone (using the best configurations on the dev set).

| Model | Exact match | Sentence BLEU | Corpus BLEU |
|---|---|---|---|
| Baseline | 0.152 | 0.743 | 0.723 |
| PATOIS | **0.197** | **0.780** | **0.766** |

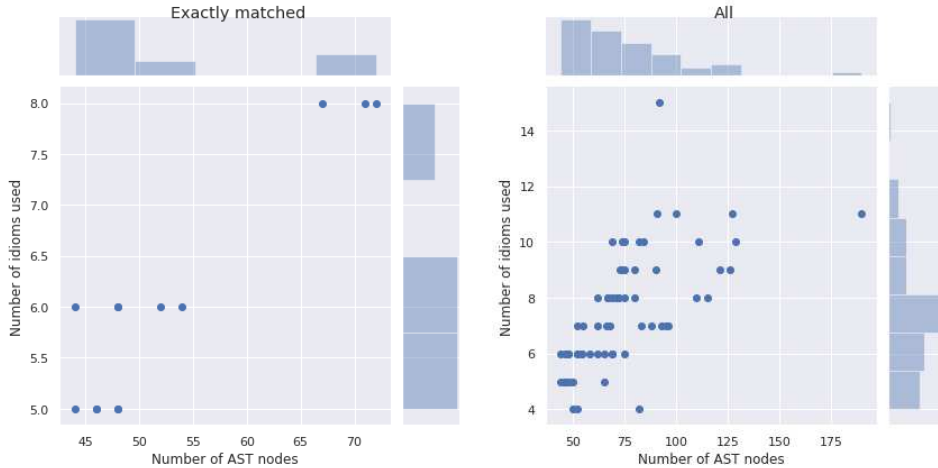

```
def __init__(self):
    super().__init__( ℓ₀: str , ℓ₁: int ,
        CHARACTER_CLASS. ℓ₃: id ,
        CARD_RARITY. ℓ₄: id , ℓ₅? )
```

```
ℓ₀: id =
    copy.copy( ℓ₁: expr )
```
```
class ℓ₀: id ( ℓ₁: id ):
    def __init__(self):
```

```
SELECT COUNT( ℓ₀: col ), ℓ₁*  WHERE ℓ₂*
INTERSECT ℓ₄?: sql EXCEPT ℓ₅?: sql
```
```
WHERE ℓ₀: col = $terminal
```

Figure 4: Five examples of commonly used idioms from the Hearthstone and Spider datasets.

Figure 5: The distribution of used idioms in the inferred ASTs on the Hearthstone test set. *Left:* in the ASTs exactly matched with ground truth; *Right:* all ASTs.

generalize well to the whole data distribution. As mentioned above, we compare only to the same baseline architecture for fairness, but PATOIS could also be easily implemented on top of the structural CNN decoder of Sun et al. [35], the current state of the art on the Hearthstone dataset.

Figure 4 shows some examples of idioms that were frequently used by the model. On Hearthstone, the most popular idioms involve common syntactic elements (*e.g.* class and function definitions) and domain-specific APIs commonly used in card implementations (*e.g.* CARD_RARITY enumerations or copy.copy calls). On Spider, they capture the most common combinations of SQL syntax, such as a SELECT query with a single COUNT column and optional INTERSECT or EXCEPT clauses. Notably, popular idioms are also often *big*: for instance, the first idiom in Figure 4 expands to a tree fragment with more than 20 nodes. Emitting it in a single step vastly simplifies the decoding process.

We further conducted qualitative experiments to analyze actual idiom usage by PATOIS on the Hearthstone test set. Figure 5 shows the distribution of idioms used in the inferred (not ground truth) ASTs. A typical program involves 7 idioms on average, or 6 for the programs that exactly match the ground truth. Despite the widespread usage of idioms, not all of the mined idioms $\widetilde{\mathcal{I}}$ were useful: only 51 out of $K = 80$ idioms appear in the inferred ASTs. This highlights the need for an end-to-end version of PATOIS where idiom mining would be directly optimized to benefit synthesis.

## 6   Related Work

**Program synthesis & Semantic parsing**   Program synthesis from natural language and input-output examples has a long history in Programming Languages (PL) and Machine Learning (ML) communities (see Gulwani et al. [15] for a survey). When an input specification is limited to natural language, the resulting problem can be considered *semantic parsing* [22]. There has been a lot of recent interest in applying recurrent sequence-based and tree-based neural networks to semantic parsing [11, 18, 21, 38, 40]. These approaches commonly use insights from the PL literature, such as grammar-based constraints to reduce the search space, non-deterministic training oracles to enable multiple executable interpretations of intent, and supervision from program execution. They typically either supervise the training on one or more golden programs, or use reinforcement learning to supervise the training from a neural program execution result [26]. Our PATOIS approach is

applicable to any underlying neural semantic parsing model, as long as it is supervised by a corpus of golden programs. It is, however, most easily applicable to tree-based and graph-based models, which directly emit the AST of the target program. In this work we have evaluated PATOIS as applied on top of the sequence-to-tree decoder of Yin and Neubig [38], and extended it with a novel training regime that teaches the decoder to emit idiom operators in place of the idiomatic code fragments.

**Sketch generation**  Two recent works [12, 25] learn abstractions of the target program to compress and abstract the reasoning process of a neural synthesizer. Both of them split the generation process into *sketch generation* and *sketch completion*, wherein the first stage emits a partial tree/sequence (*i.e.* a *sketch* of the program) and the second stage fills in the holes in this sketch. While sketch generation is typically implemented with a neural model, sketch completion can be either a different neural model or a combinatorial search. In contrast to PATOIS, both works define the grammar of sketches manually by a deterministic *program abstraction* procedure and only allow a single top-level sketch for each program. In addition, an earlier work of Bošnjak et al. [6] also formulates program synthesis as sketch completion, but in their work program sketches are manually provided rather than learned. In PATOIS, we learn the abstractions (code idioms) automatically from a corpus and allow them to appear anywhere in the program, as is common in real-life programming.

**Learning abstractions**  Recently, Ellis et al. [13] developed an Explore, Compress & Compile ($EC^2$) framework for automatically learning DSLs for program synthesis from I/O examples (such as the DSLs used by FlashFill [14] and DeepCoder [5]). The workflow of $EC^2$ is similar to PATOIS, with three stages: **(a)** learn new DSL subroutines from a corpus of tasks, **(b)** train a recognition model that maps a task specification to a distribution over DSL operators as in DeepCoder [5], and **(c)** use these operators in a program synthesizer. PATOIS differs from $EC^2$ in three aspects: **(i)** we assume a natural language specification instead of examples, **(ii)** to handle NL specifications, our synthesizer is a neural semantic parser instead of enumerative search, and **(iii)** most importantly, we discover idioms that compress general-purpose languages instead of extending DSLs. Unlike for inductive synthesis DSLs such as FlashFill, the existence of *useful* DSL abstractions for general-purpose languages is not obvious, and our work is the first to demonstrate them.

Concurrently with this work, Iyer et al. [17] developed a different approach of learning code idioms for semantic parsing. They mine the idioms using a variation of *byte-pair encoding* (BPE) compression extended to ASTs and greedily rewrite all the dataset ASTs in terms of the found idioms for training. While the BPE-based idiom mining is more computationally efficient than non-parametric Bayesian inference of PATOIS, introducing ASTs greedily tends to lose information about overlapping idioms, which we address in PATOIS using our novel training objective described in Section 4.2.

As described previously, our code idiom mining is an extension of the procedure developed by Allamanis et al. [2, 3]. They are the first to use the tree substitution grammar formalism and Bayesian inference to find non-trivial common idioms in a corpus of code. However, their problem formalization does not involve any application for the learned idioms beyond their explanatory power.

# 7   Conclusion

Semantic parsing, or neural program synthesis from natural language, has made tremendous progress over the past years, but state-of-the-art models still struggle with program generation at multiple levels of abstraction. In this work, we present a framework that allows incorporating learned coding patterns from a corpus into the vocabulary of a neural synthesizer, thus enabling it to emit high-level or low-level program constructs interchangeably at each generation step. Our current instantiation, PATOIS, uses Bayesian inference to mine common code idioms, and employs a novel nondeterministic training regime to teach a tree-based generative model to optionally emit whole idiom fragments. Such dataset abstraction using idioms improves the performance of neural program synthesis.

PATOIS is only the first step toward learned abstractions in program synthesis. While code idioms often correlate with latent semantic concepts and our training regime allows the model to learn which idioms to use and in which context, our current method does not mine them with the intent to directly optimize their usefulness for generation. In future work, we want to alleviate this by jointly learning the mining and synthesis models, thus optimizing the idioms' usefulness for synthesis by construction. We also want to incorporate *program semantics* into the idiom definition, such as data flow patterns or natural language phrases from task specs.

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
