[Reviews · NeurIPS 2019]

Reviewer 1



Summary: This paper proposes a semantic parsing and program synthesis method. Code generation relies on low-level and high-level abstractions. High-level abstractions can be thought of as functions that are re-used in several programs. In order to model high-level abstraction, the authors propose using a code-idiom mining method from the literature. Once the code idioms are extracted, the program is generated. The generative process has the capability of spitting tokens or idioms. Strengths: The paper is well-written and well organized. The motivation is stated clearly and contrasted well against the literature. The experiments seem satisfying, although I have a follow-up question that I will state in the next section. The proposed model is novel and seems to be suitable for the problem at hand. However, I do have a follow-up question about the proposed two-step model that I will state in the next section. Weaknesses: I did not understand what the baseline is. How does the decoder in section 4 different from the proposed model? Also, one would expect to see the performance compared to other similar methods stated by the authors [12,23] that rely on sketch generation. Since this proposed model is closest in the high-level idea to these methods. How is the idiom mining part of the model evaluated? It seems like the idiom mining step introduces a point of failure into the model that cannot be trained end-to-end. Does the authors have any comments regarding this? Have they performed any analysis of the possible failures that are a result of this step? This reference seems to be missing: Riedel, Sebastian, Matko Bosnjak, and Tim Rocktäschel. "Programming with a differentiable forth interpreter." CoRR, abs/1605.06640 (2016). It learns program sketches and then fills in the sketch. --------------------------------------- Post rebuttal update: I have read the response and maintain my accept rating of 7.

Reviewer 2



The paper is written well and easy to follow. Interesting iand novel idea to use code-idioms for program synthesis tasks. Mining the idioms use existing bayesian non-paaraa The paper highlights its advantage over other sketch generation works, due to its capability to generalize with few code fragments instead of grammatically correct complete intermediate template. Experimental analysis comparing the two would have provided more insights. The time complexity especially of idiom miner step should be added.

Reviewer 3



This paper proposes a new framework which utilizes idiom mining to improve existing models for semantic parsing. The authors argument that one of the main challenges of synthesizing a program is the insufficient separation between the high-level and low-level reasoning, which forces the model having to learn both the common high-level abstractions and the low-level details that specify the semantic meaning of the generated program. They propose to frame idiom mining as a nonparametric Baysian problem and first learn a set of code idioms, which are AST fragments of the programs, from the training set. The mined idioms are then used to train the generation model which teaches the model to make use of common code snippets and not just generating the program by predicting the AST node by node. The main idea presented in the paper appears to be interesting and insightful, particularly for semantic parsing where access to parallel corpus is often limited. The paper is clearly presented and relatively easy to follow. Experiment results on two benchmark datasets Hearthstone and Spider show noticeable improvements over the selected baselines when the proposed method is used. Although no results on the current state-of-the-art were mentioned in the paper.

[Author Response · NeurIPS 2019]

We would like to thank the reviewers for their feedback. We will add the suggested references, clarifications from our
answers below, and further qualitative experiments in the camera-ready revision.

**[R1] Which parts of the model in §4 constitute the baseline?** The AST-based decoder described in §4.1 constitutes
our baseline. Its architecture largely follows Yin and Neubig [3]. PATOIS adds two novel contributions on top of it,
described in §4.2: **(a)** the new objective in Eq. (4) that allows the model to emit idiom rules instead of original CFG
rules, and **(b)** the new training regime that teacher-forces idiom bodies when an idiom rule is chosen. We will specify
this more clearly in the camera-ready revision.

**[R1] How does PATOIS compare against methods based on sketch generation?** Coarse2Fine and Bayou both
require a manually-defined formulation of program sketches unlike PATOIS which learns idioms from data. There only
exists one sketch per program, unlike idioms which can occur at multiple places within the program. Fig. 1 shows
that when run on the Hearthstone test set, PATOIS invokes 4–15 idioms in the course of decoding. Moreover, larger
programs use more idioms, showing that the decoder of PATOIS learns to switch between high-level and low-level
reasoning repeatedly, rather than learning only high-level sketches.

**[R1] What are the possible failure modes of idiom mining?** The most important failure mode of idiom mining
is proposing idioms in $\widetilde{\mathcal{I}}$ that end up unused by the synthesis model despite being common. For instance, our best
Hearthstone model never used 29 out of $K = 80$ idioms on the test set despite them matching in some ASTs. Another
possible failure is overfitting idioms to the training set. We tackle it by filtering out idioms that do not occur in the
validation set, and empirically demonstrate that the remaining ones generalize to the test set. In general, we evaluate
$\widetilde{\mathcal{I}}$ as a hyperparameter, choosing the vocabulary that optimizes validation performance (as in Tables 2-3). We have
ongoing work on an end-to-end extension of PATOIS, but this extension is beyond the scope of this work.

**[R2] What is the runtime complexity?** PATOIS has two phases: (1) idiom mining and (2) training the synthesis model.
We outline the complexity analysis for them below, and will add a complete one to the camera-ready revision.

Phase 1 implements MCMC sampling, run for $M = 10$ iterations. At each iteration, PATOIS traverses each AST $T \in \mathcal{D}$
once to sample the random variables that partition it into the idiom fragments (see §3). Thus, the complexity of mining
is $\mathcal{O}(M \cdot \sum_{T \in \mathcal{D}} |T|)$. In practice, for the 10,181 ASTs in Spider it took $< 30$ min on a 32-core 2.4 GHz Intel Xeon®.

Phase 2 has essentially the same complexity as the baseline training thanks to our objective in Eq. (4). For a training
instance $\langle X, T \rangle$ computing the loss takes $\mathcal{O}(|T|)$ time. Each step computes cross-entropy between the predicted
distribution over production rules and the one-hot distribution with the ground truth rule. In Eq. (4), the cross-entropy
now allows $> 1$ ground truth rules: the original CFG rule and any matching idioms. The asymptotic complexity of
cross-entropy is the same, so the overall per-instance complexity remains at $\mathcal{O}(|T|)$; however, there are more production
rules involved, which increases the cost of computing the predicted distribution itself.

**[R1, R3] Is the method applicable to state-of-the-art models? What would be the improvement?** At the time of
writing, the state of the art on Hearthstone and Spider is achieved by GrammarCNN [2] and IRNet [1], respectively.
Notably, both of them (like many other contemporary models) use *structural AST-based decoders*, trained using the
cross-entropy objective over the AST production rules. As we describe in §4, the PATOIS framework is applicable to
any decoder that follows such architecture. We only compared against our baseline for fairness, but may be able to also
implement PATOIS on top of the open-sourced GrammarCNN for the camera-ready revision.

The improvement of PATOIS should benefit any such structural decoder because idioms fundamentally can help to *avoid*
*mistakes in modeling the generation of idiom bodies* (which account for a sizable fraction of the AST). The effect will be
less prominent for IRNet where the Coarse2Fine sketching mechanism par-
tially accomplishes the same goal, and more prominent for GrammarCNN.

**[R2, R3] Please provide more qualitative experiments. How often are**
**the idioms used?** We agree that §5 needs more qualitative experiments,
and will add them to the camera-ready revision. We conducted some during
the author response period to supplement our answers. For instance, Fig. 1
shows a distribution of idiom usage on the Hearthstone test set.

[1] J. Guo, Z. Zhan, Y. Gao, Y. Xiao, J.-G. Lou, T. Liu, and D. Zhang. Towards
complex text-to-SQL in cross-domain database with intermediate representation.
In *ACL*, July 2019.

[2] Z. Sun, Q. Zhu, L. Mou, Y. Xiong, G. Li, and L. Zhang. A grammar-based
structural CNN decoder for code generation. In *AAAI*, 2019.

[3] P. Yin and G. Neubig. A syntactic neural model for general-purpose code
generation. In *ACL*, July 2017.

Figure 1: Hearthstone idiom usage on
the test set. Number of idioms used and
AST nodes are from synthesized trees, not
ground truth.

[Meta-Review · NeurIPS 2019]

The authors should be commended for writing and submitting a solid paper on semantic parsing and program synthesis that was clearly written, deemed interesting, and contained good and compelling experimental results. An accept.